# Neural Star Domain as Primitive Representation

**Yuki Kawana**[1], **Yusuke Mukuta**[1,2], **Tatsuya Harada**[1,2]
[1]The University of Tokyo, [2]RIKEN AIP
{kawana, mukuta, harada}@mi.t.u-tokyo.ac.jp

## Abstract

Reconstructing 3D objects from 2D images is a fundamental task in computer vision, and an accurate structured reconstruction by parsimonious and semantic primitive representations has an even broader application range. When a target shape is reconstructed using multiple primitives, it is preferable that its fundamental properties, such as collective volume and surface, can be readily and comprehensively accessed so that the primitives can be treated as if they were a single shape. This becomes possible by a primitive representation with unified implicit and explicit representations. However, primitive representations in current approaches do not satisfy these requirements. To resolve this, we propose a novel primitive representation termed neural star domain (NSD) that learns primitive shapes in a star domain. We demonstrate that NSD is a universal approximator of the star domain; furthermore, it is not only parsimonious and semantic but also an implicit and explicit shape representation. The proposed approach outperforms existing methods in image reconstruction tasks in terms of semantic capability as well as sampling speed and quality for high-resolution meshes.

## 1 Introduction

Understanding 3D objects by decomposing them into simpler shapes (termed primitives) has been widely studied in computer vision [1–3]. Decomposing 3D objects into parsimonious and semantic primitive representations is important for understanding their structure. Constructive solid geometry [4] uses combinations of primitives to reconstruct complex shapes.

Recently, learning-based approaches have been adopted to primitive based approaches [5–11]. It has been demonstrated that these approaches enable a semantically consistent part arrangement in various shapes. Moreover, the use of implicit representations allows the set of primitives to be represented as a single collective shape by considering a union [5, 6, 12]; this can improve the reconstruction accuracy during training.

However, the expressiveness of primitives, particularly those with closed shapes, has been limited to simple shapes (cuboids, superquadrics, and convex shapes). Although primitives can learn semantic part arrangements, the semantic shapes of the parts cannot be learned using existing methods. In addition, although the union of primitive volumes could be represented by implicit representations in previous studies, the lack of immediate access to the union of primitive surfaces during training results in complex training schemes [5, 6, 12].

It is challenging to define a primitive that addresses all these problems. State-of-the-art expressive primitives with explicit surfaces do not have implicit representations [13, 7], and thus they cannot efficiently consider unions of primitives to represent collective shapes. Leading primitive representations by convex shapes [5, 6] with implicit representations involve a tradeoff regarding the number $H$ of half-space hyperplanes defining a convex. Using more hyperplanes yields more expressive convex shapes at the expense of a quadratically growing computation cost in extracting differentiable

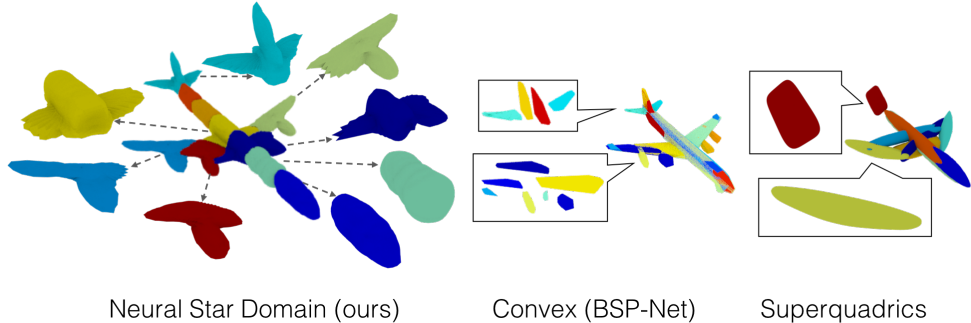

Neural Star Domain (ours)        Convex (BSP-Net)        Superquadrics

Figure 1: Overview of proposed approach. The primitives have a more meaningful and wider shape variety compared with those in previous studies.

|  | Implicit | Explicit | Semantic | Parsimonious | Accurate |
|---|---|---|---|---|---|
| DMC [14] | ✓ | ✓ | – | – | ✓ |
| SQ [8] |  | ✓ |  | ✓ |  |
| AtlasNetV2 [7] |  | ✓ | ✓ | ✓ | ✓ |
| BSP-Net [5] | ✓ |  |  |  | ✓ |
| Ours | ✓ | ✓ | ✓ | ✓ | ✓ |

Table 1: Overview of shape representations in previous studies. SQ stands for superquadrics [8]. We regard a primitive as having an explicit representation if it can access the explicit surface in *both* the inference and the training process. Moreover, a primitive representation is said to be semantic if it can reconstruct semantic *shapes* in addition to part correspondence.

surface points. A naive implementation costs $O(H^2)$ to filter the surface points of a convex from the hyperplanes.

To address these issues, we propose a novel primitive representation termed neural star domain (NSD) that learns shapes in a star domain by using neural networks. A star domain is a group of arbitrary shapes that can be represented by a continuous function defined on the surface of a sphere. As it can express concavity, we can regard it as a generalized shape representation of convex shapes. The learned primitives are visualized in Figure 1. Moreover, we can directly approximate star-domain shapes using neural networks owing to their continuity. We demonstrate that the complexity of the shapes that can be represented by an NSD is equivalent to the approximation ability of the neural network. In addition, as it is defined on the surface of a sphere, a primitive can be represented in both implicit and explicit forms by transforming it between spherical and Cartesian coordinates. The proposed approach is compared with those in previous studies in Table 1.

The contributions of this study can be summarized as follows: (1) We propose a novel primitive representation with high expressive power, and we demonstrate that it is more parsimonious and can learn semantic part shapes. (2) We demonstrate that the proposed primitive provides unified implicit and explicit representations that can be used during training and inference, leading to improved mesh reconstruction accuracy and speed.

## 2 Related work

Methods for decomposing shapes to primitives have been studied extensively in computer vision [1]. Some of the classical primitives used in computer vision are generalized cylinders [2] and geons [3]. In deep generative models, cuboids [11, 10] and superquadrics [8, 9] are used to realize consistent parsing across shapes. However, these methods have poor reconstruction accuracy owing to the limitations in the parameter spaces of the primitives. Thus, their application is limited to shape abstraction. Using parametrized convex shapes for improved reconstruction accuracy has been recently proposed in [5, 6]. However, as the shapes of the primitives are constrained to be convex, their interpretability is limited to part parsing. In this study, we investigate star domains as primitive representations with more expressive power than that of previously proposed primitive representations.

In computation theory, 2D polygonal shape decomposition using star domains has a long history [15, 16]. In computer vision, star domains have been used to abstract 3D shapes to *encode* shape embedding [17–20] for discriminative models. In contrast, we study the application of star domains to *decode* shape embedding to accurately reconstruct 3D shapes for generative models.

Surface representation of 3D objects in the context of generative models has been studied extensively. In recent studies, the standard explicit shape representation for generative models is a mesh [13, 21–23, 13]. Meshes [24], pointclouds [7], and parametrized surfaces [11, 10, 8, 9] have been studied as explicit surfaces for primitive models. A state-of-the-art method employs a learnable indicator function for non-primitive- [25, 26] and primitive-based approaches [12, 5, 6]. However, extracting a surface mesh during inference is quite costly, as the isosurface extraction operation grows cubically for the desired meshing resolutions. An implicit representation model with fast polymesh sampling during inference was proposed in [5]. However, owing to the lack of explicit surface representations during training, primitive-based methods with implicit representations require complicated training schemes, such as near-surface training data sampling with ray casting [12, 6], and heuristic losses to keep primitives inside the shape boundary [6], or a multi-stage training strategy to approximate explicit surfaces [5]. A notable exception that uses both implicit and explicit representations was proposed in [14]; however, this is possible by reconstructing the shape as a voxel at the cost of limited shape resolution. In this study, we propose a unified shape representation in both explicit and implicit forms at an arbitrary resolution. This is used to realize a simple training scheme with fast high-resolution mesh sampling during inference.

## 3 Methods

We first formulate the problem setting in Section 3.1. Subsequently, we define star domains in Section 3.2. In addition, we introduce NSDs to approximate shapes in star domains, with a theoretical analysis of the representation power. Using NSDs as building blocks, we describe the pipeline of the proposed approach in Sections 3.3, 3.4, and 3.5. Implementation details are provided in Section 3.6.

### 3.1 Problem setting

We represent an object shape as a set of surface points $P \subseteq \mathbb{R}^3$, and as an indicator function that can be evaluated at an arbitrary point $\mathbf{x} \in \mathbb{R}^3$ in 3D space as $O : \mathbb{R}^3 \to \{0, 1\}$, where $\{\mathbf{x} \in \mathbb{R}^3 \,|\, O(x) = \tau\}$. In this equation, $\tau = 0$ defines the outside of the object, and $\tau = 1$ defines the inside. Our objective is to parametrize the 3D shape by a composite indicator function $\hat{O}$ and surface points $\hat{P}$ that can be decomposed into a collection of $N$ primitives. The $i$th primitive has an indicator function $\hat{O}_i : \mathbb{R}^3 \to [0, 1]$ and a surface point function defined on a sphere $\hat{P}_i : \mathbb{S}^2 \to \mathbb{R}^3$. To realize implicit and explicit shape representation simultaneously, we further require $\hat{O}$ and $\hat{P}$ to be related as $\hat{O}(\hat{p}) = \tau_o$, where $\hat{p} \in \hat{P}$, and $\tau_o \in [0, 1]$ is a constant that represents the isosurface. We ensure that both the composite indicator function and the surface points are approximated as $O \approx \hat{O}$ and $P \approx \hat{P}$, respectively, through training losses.

### 3.2 Neural star domain

A geometry $U \subseteq \mathbb{R}^3$ is a star domain if $\exists \mathbf{t} \in U, \forall \mathbf{u} \in U, [\mathbf{t}, \mathbf{u}] = \{(1 - v)\mathbf{t} + v\mathbf{u}, 0 \le v \le 1\} \subseteq U$. Intuitively, a star domain is any geometry with an origin $\mathbf{t}$ such that a straight line segment between any point $\mathbf{u}$ inside the geometry and $\mathbf{t}$ is also inside the geometry. Thus, we can regard star domain shapes as continuous functions defined on the surface of a sphere. We denote such functions as $r : \mathbb{S}^2 \to \mathbb{R}$. The spherical harmonics expansion $\mathbb{S}^2 \to \mathbb{R}$ is a multivariate polynomial function that is also defined on the surface of a sphere. Thus, we can formulate a star domain using a spherical harmonics expansion as

$$r(\mathbf{d}) = \sum_{l=0}^{\infty} \sum_{m=-l}^{l} c_{l,m} Y_{l,m}(\omega(\mathbf{d})), \ \omega(\mathbf{d}) = (\sin\theta\cos\phi, \sin\theta\sin\phi, \cos\theta), \quad (1)$$

where $\mathbf{d} = (\theta, \phi) \in \mathbb{S}^2$, $c_{l,m} \in \mathbb{R}$ is a constant, and $Y_{l,m}$ is the Cartesian spherical harmonic function [27]. Examples of $Y_{l,m}$ given $(l, m)$ can be found in Appendix.

To realize the star domain primitive, we propose an NSD, which approximates $r$ by a neural network $f_{NN}$, taking $\omega(\cdot)$ as input.

**Approximation ability.**   We demonstrate the universal approximation ability of the NSD to a star domain $r$. The following theorem implies that $r$ can be arbitrarily approximated by an NSD.

**Theorem.** Let $r : \mathbb{S}^2 \to \mathbb{R}$ be a continuous function on the surface of a sphere. Then, $\forall \epsilon > 0$, $\exists$ an NSD $f_{NN} \circ \omega : \mathbb{S}^2 \to \mathbb{R}$ such that for any $\mathbf{d} \in \mathbb{S}^2$, we have

$$|r(\mathbf{d}) - f_{NN}(\omega(\mathbf{d}))| < \epsilon. \tag{2}$$

*Proof.* By the completeness of spherical harmonics [28] to a continuous function on a spherical surface, as shown in Equation (1), $\forall \epsilon_1 > 0$, $\exists L \in \mathbb{N}^+$ and $c_{l,m} \in \mathbb{R}$ such that for any $\mathbf{d} \in \mathbb{S}^2$, we have

$$|r(\mathbf{d}) - r_L(\mathbf{d})| < \epsilon_1, \text{ where } r_L(\mathbf{d}) = \sum_{l=0}^{L} \sum_{m=-l}^{l} c_{l,m} Y_{l,m}(\omega(\mathbf{d})). \tag{3}$$

$\omega$ can be regarded as $Y_{1,m}$ with an appropriate constant $c_{1,m}$, and from the definition of Cartesian spherical harmonics [27], each $Y_{l,m}$ with $l > 1$ can be written as a polynomial function of $Y_{1,m}$ with an appropriate constant $c_{l,m}$. Thus, $r_L$ can be regarded as a polynomial function over $\omega$, that is, it is continuous over $\omega$.

By the universal approximation theorem of neural networks to a continuous function [29, 30], $\forall \epsilon_2 > 0$, $\exists$ a neural network $f_{NN} : \mathbb{R}^3 \to \mathbb{R}$ such that for any $\omega_{\mathbf{d}} \in \{\omega(\mathbf{d}) \,|\, \mathbf{d} \in \mathbb{S}^2\}$, we have

$$|r_L(\mathbf{d}) - f_{NN}(\omega_{\mathbf{d}})| < \epsilon_2. \tag{4}$$

Given Equations (3) and (4), $\forall \epsilon > 0$, $\exists$ a neural network $f_{NN} : \mathbb{R}^3 \to \mathbb{R}$ such that for any $\mathbf{d} \in \mathbb{S}^2$, we have

$$|r(\mathbf{d}) - f_{NN}(\omega(\mathbf{d}))| < \epsilon_1 + \epsilon_2 = \epsilon. \tag{5}$$

$\square$

It should be noted that there exist network architectures that take the output values of trigonometric functions as input, such as HoloGAN [31]. However, the proposed approach differs in the input and output as follows: (1) By taking $\omega$ as input, the proposed approach is theoretically founded on an approximate spherical harmonic expansion. HoloGAN takes the output values of high-degree trigonometric polynomial functions as input. (2) The neural network in HoloGAN is aimed at predicting high-dimensional vectors as images, whereas the proposed approach is aimed specifically at predicting a single-dimensional radius $r$ by approximating the star domain.

### 3.3   Primitive representation

As an NSD is defined on the surface of a sphere, one can define both implicit and explicit shape representations of a primitive. For simplicity, we define an NSD $f := f_{NN} \circ \omega$ in the following sections.

**Implicit representation.**   Given the 3D location $\mathbf{x} \in \mathbb{R}^3$, an indicator function $\hat{O}_i : \mathbb{R}^3 \to [0, 1]$ for the $i$th primitive located at $\mathbf{t}_i$ is expressed as follows:

$$\hat{O}_i(\mathbf{x}; \mathbf{t}_i) = \text{Sigmoid}(\alpha(1 - \frac{\|\bar{\mathbf{x}}\|}{r^+})), \text{ where } \bar{\mathbf{x}} = \mathbf{x} - \mathbf{t}_i, \; r^+ = \text{ReLU}(f_i(G(\mathbf{x}))), \tag{6}$$

where $\alpha$ is a scaling factor that adjusts the margin of the indicator values between the inside and outside of the shape, $G : \mathbb{R}^3 \to \mathbb{S}^2$ denotes the conversion from 3D Cartesian coordinates to the spherical surface, and the ReLU operator ensures that the estimated radius is a non-negative real number. We note that $\|\bar{\mathbf{x}}\| - r^+$ can be interpreted as a singed distance function. The formulas for $G$ and $G^{-1}$ can be found in Appendix.

**Explicit representation.**   With a slight abuse of notation, we denote the conversion from spherical coordinates to a 3D location as $G^{-1} : \mathbb{R} \times \mathbb{S}^2 \to \mathbb{R}^3$. We can sample a surface point in the direction of $\mathbf{d}$ from the origin of the $i$th primitive located at $\mathbf{t}_i$ as follows:

$$\hat{P}_i(\mathbf{d}; \mathbf{t}_i) = G^{-1}(r^+, \mathbf{d}) + \mathbf{t}_i, \text{ where } r^+ = \text{ReLU}(f_i(\mathbf{d})). \tag{7}$$

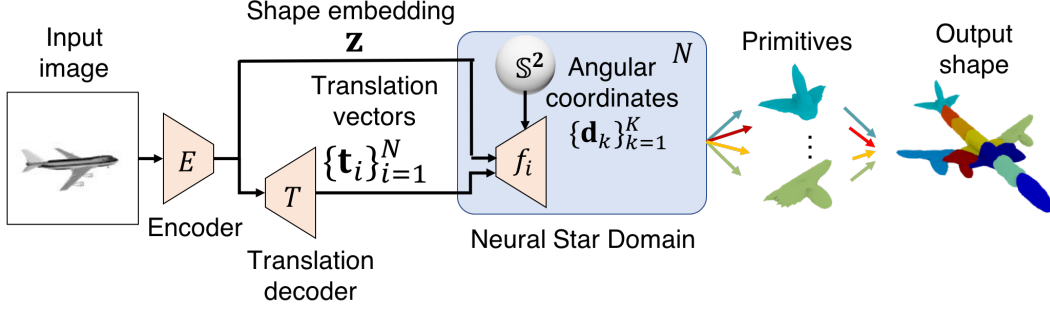

Figure 2: Architecture of NSDN.

## 3.4 Neural star domain network

To represent the target shape as a collection of primitives, we define an NSD Network (NSDN), which employs a bottleneck auto-encoder architecture similar to that in [25]. An NSDN consists of an encoder $E$, a translation network $T$, and a set of NSDs $\{f_i\}_{i=1}^N$. Given an input $I$, the encoder $E$ derives a shape embedding $\mathbf{z}$. Then, the translation network $T$ outputs a set of translation vectors $\{\mathbf{t}_i\}_{i=1}^N$ from $\mathbf{z}$. The translation vectors represent the location of each primitive. The $i$th NSD $f_i$ acts as a decoder and infers the radius given an angular coordinate $\mathbf{d}$, translation vectors $\mathbf{t}_i$, and a shape embedding $\mathbf{z}$. In this study, we only estimate the location as the pose of the primitives, whereas in previous studies, the scale and rotation of each primitive were additionally predicted [6, 8, 11]. We observe that learning the rotation and scale leads to unsuccessful training. An overview of the architecture is shown in Figure 2.

**Composite indicator function.** To derive an implicit representation of the NSDN, we define a composite indicator function as the union of $N$ NSD indicator functions as

$$\hat{O}(\mathbf{x}; \{\mathbf{t}_i\}_{i=1}^N) = \text{Sigmoid}(\sum_{i \in [N]} \hat{O}_i(\mathbf{x}; \mathbf{t}_i)). \tag{8}$$

To encourage gradient learning of all primitives during training, we use the sum of the indicator values over the primitives rather than the maximum value. We treat the threshold of the indicator value $\tau_o$ of the surface level of $\hat{O}$ as a hyperparameter.

**Surface point extraction.** Owing to the unified explicit and implicit shape representation of the NSD, the NSDN can extract the union of the surface points of the primitives in a differentiable manner. We define the unified surface points as follows:

$$\hat{P} = \bigcup_i \{\hat{P}_i(\mathbf{d}; \mathbf{t}_i) | \forall j \in [N \setminus i], \hat{O}_j(\hat{P}_i(\mathbf{d}; \mathbf{t}_i), \mathbf{t}_i) < \tau_s, \mathbf{d} \in \{\mathbf{d}_k\}_{k=1}^K\}, \tag{9}$$

where $K$ denotes the number of points sampled from the surface of the sphere, and $\tau_s$ is a hyperparameter for the threshold of the indicator value for the surface points.

## 3.5 Training loss

To learn the parameters $\mathbf{\Theta}$ of the NSDN, we define the *surface point loss*, which minimizes the symmetric chamfer distance between the surface points $P$ from a training sample and those from the predicted surface points $\hat{P}$. The surface point loss is formulated as

$$L_S(\mathbf{\Theta}) = \mathbb{E}_{\hat{p} \sim \hat{P}} \min_{p \sim P} \|\hat{p} - p\| + \mathbb{E}_{p \sim P} \min_{\hat{p} \sim \hat{P}} \|p - \hat{p}\|. \tag{10}$$

We note that the surface point loss enables learning collective surfaces of primitives by accessing both implicit and explicit representations, as shown in Equation 9. The training loss leads to a better reconstruction than minimizing the distance between $P$ and a simple union of the surface points of the primitives $\bigcup_{i \in [N]} \{\hat{P}_i(\mathbf{d}; \mathbf{t}_i) | \mathbf{d} \in \{\mathbf{d}_k\}_{k=1}^K\}$, as in [8, 11]. This is because, ideally, the loss

should measure the distance between the two sets of surface points. We also use the occupancy loss as in [25] $L_O(\boldsymbol{\Theta}) = \mathbb{E}_{\mathbf{x} \sim \mathbb{R}^3} \text{BCE}(O(\mathbf{x}), \hat{O}(\mathbf{x}))$, where BCE is the binary cross entropy. We observe that using the occupancy loss in addition to the surface point loss achieves the best reconstruction performance.

## 3.6 Implementation details

In all experiments, we use the same architecture, whereas the number of primitives $N$ varies. $N$ is set to 30 by default, unless stated otherwise. We use ResNet18 as the encoder $E$, which produces shape embedding as a latent vector $\mathbf{z} \in \mathbb{R}^{256}$ for an input RGB image by following OccNet [25]. For the translation network $T$, we use a multilayer perceptron (MLP) with three hidden layers with $(128, 128, N * 3)$ units with ReLU activation. For an NSD, we use an MLP with three hidden layers with $(64, 64, 1)$ units and ReLU activation. We set the margin $\alpha$ of the indicator function to 100. The threshold $\tau_o$ of the composite indicator function is determined by a grid search over the validation set. For example, for $N = 30$, we use $\tau_o = 0.99$. We use 0.1 for the threshold $\tau_s$ of surface point extraction. During training, we use a batch size of 20, and train with the Adam optimizer, with a learning rate of 0.0001. We set the weight of $L_o$ and $L_s$ as 1 and 10, respectively. For the training data, we sample 4096 points from the ground-truth pointcloud, and $400 * N$ samples from the generated shape for the surface point loss $L_s$; moreover, we sample 2048 points from the ground-truth indicator values for the indicator loss $L_o$. For mesh sampling, we use a spherical mesh template.

## 4 Experiments

**Dataset.** In the experiments, we use the ShapeNet [32] dataset. Following [25], we test the proposed approach on 13 categories of objects. In addition, we use the same samples and data split as in [25]. For 2D images, we use the rendered view provided in [33]. For the quantitative evaluation of the part semantic segmentation, we use PartNet [34] and the part labels provided in [35].

**Methods.** We compare the proposed approach with several state-of-the-art approaches using different shape representations. Regarding primitive-based reconstruction approaches, we compare the proposed method with BSP-Net [5], CvxNet [6], and SIF [12] (implicit-representation), and with AtlasNetV2 [7] (explicit representation). As the approaches in [5, 6] represent shapes as collections of convex shapes, we regard them as a baseline for the effectiveness of the star-domain primitive representation. Regarding non-primitive-based reconstruction approaches, we compare the proposed method with OccNet [25], which is the leading implicit representation technique, and with AtlasNet [13] (explicit shape representation). Concerning AtlasNetV2, as the code provided by the author does not include a model for single-view reconstruction, we replace the provided encoder with the same ResNet18 used by NSDN and OccNet, and train the model. Furthermore, for a fair comparison with NSDN, we sample 400 points from each patch during training, and use 30 patches for AtlasNetV2, unless otherwise noted. We confirm that this leads to a slightly better reconstruction accuracy than the original configuration. For BSP-Net, we use the pretrained model described in Section 4.1. In Section 4.2, we train BSP-Net using the code provided in [5]. As BSP-Net uses different train and test splits, we evaluate it on the intersection of the test splits from [25] and [5].

**Metrics.** We evaluate the proposed methods in terms of reconstruction accuracy, part correspondence, and mesh sampling speed. To evaluate the reconstruction accuracy, we apply three commonly used metrics to compute the difference between the reconstruction meshes and the ground truth: (1) F-score, which, by the argument in [36], can be interpreted as the percentage of correctly reconstructed surfaces, (2) L1 chamfer distance (CD1), and (3) volumetric IoU (IoU). For all metrics, we use 100,000 sample points from the ground-truth meshes, and reconstruct shape meshes by following [25, 6]. To evaluate the part correspondence in semantic capability, we use the standard label IoU between the ground-truth part label and the predicted part label. Regarding mesh sampling speed, we measure the time in which a pipeline encodes an image and decodes mesh vertices and faces. We exclude the time for the device I/O. All speed measurements are performed on an NVIDIA V100 GPU. Moreover, for a fair comparison, we measure the time to mesh a single primitive for AtlasNet, AtlasNetV2, and BSP-Net analogously with parallel processing, because their original implementation sequentially processes each primitive for meshing, whereas ours does meshing is parallel.

| | | airplane | bench | cabinet | car | chair | display | lamp | speaker | rifle | sofa | table | phone | vessel | mean | time |
|---|---|---|---|---|---|---|---|---|---|---|---|---|---|---|---|---|
| F-score | AtlasNet [13] | 67.24 | 54.50 | 46.43 | 51.51 | 38.89 | **42.79** | 33.04 | 35.75 | **64.22** | 43.46 | 44.93 | 58.85 | **49.87** | 48.57 | 0.008 |
| | AtlasNetV2 [7] | 54.99 | 50.67 | 31.95 | 39.73 | 29.10 | 33.55 | 28.35 | 22.54 | 62.27 | 30.15 | 45.93 | 51.45 | 39.91 | 40.05 | 0.010 |
| | OccNet [25] | 62.87 | 56.91 | 61.79 | 56.91 | 42.41 | 38.96 | 38.35 | 42.48 | 56.52 | 48.62 | 58.49 | 66.09 | 42.37 | 51.75 | 0.525 |
| | OccNet* [25] | 63.56 | 57.39 | **63.03** | 61.41 | **43.61** | 41.54 | **41.13** | **45.39** | 57.94 | **49.86** | **59.62** | 66.11 | 45.00 | 53.51 | 0.529 |
| | SIF [12] | 52.81 | 37.31 | 31.68 | 37.66 | 26.90 | 27.22 | 20.59 | 22.42 | 53.20 | 30.94 | 30.78 | 45.61 | 36.04 | 34.86 | n/a |
| | CvxNet [6] | **68.16** | 54.64 | 46.09 | 47.33 | 38.49 | 40.69 | 31.41 | 29.45 | 63.74 | 42.11 | 48.10 | 59.64 | 45.88 | 47.36 | n/a |
| | BSP-Net [5] | 61.91 | 53.12 | 44.75 | 55.24 | 38.57 | 35.68 | 29.98 | 34.04 | 57.28 | 43.89 | 46.42 | 49.18 | 42.76 | 45.60 | 0.014 |
| | NSDN (ours) | 67.96 | **60.37** | 59.26 | **63.54** | 43.58 | 41.81 | 38.83 | 43.09 | 63.31 | 48.97 | 57.91 | **70.65** | 46.49 | **54.29** | 0.014 |
| CD1 | AtlasNet [13] | 0.104 | 0.138 | 0.175 | 0.141 | 0.209 | **0.198** | **0.305** | 0.245 | 0.115 | 0.177 | 0.190 | 0.128 | **0.151** | 0.175 | 0.008 |
| | AtlasNetV2 [7] | 0.119 | 0.164 | 0.246 | 0.176 | 0.256 | 0.209 | 0.313 | 0.340 | **0.099** | 0.210 | 0.221 | 0.131 | 0.159 | 0.203 | 0.010 |
| | OccNet [25] | 0.147 | 0.155 | 0.167 | 0.159 | 0.228 | 0.278 | 0.479 | 0.300 | 0.141 | 0.194 | 0.189 | 0.140 | 0.218 | 0.215 | 0.525 |
| | OccNet* [25] | 0.141 | 0.154 | **0.149** | 0.150 | 0.206 | 0.214 | 0.369 | 0.254 | 0.142 | 0.182 | 0.175 | 0.124 | 0.194 | 0.189 | 0.529 |
| | SIF [12] | 0.167 | 0.261 | 0.233 | 0.161 | 0.380 | 0.401 | 1.096 | 0.554 | 0.193 | 0.272 | 0.454 | 0.159 | 0.208 | 0.349 | n/a |
| | CvxNet [6] | **0.093** | **0.133** | 0.160 | **0.103** | 0.337 | 0.223 | 0.795 | 0.462 | 0.106 | **0.164** | 0.358 | **0.083** | 0.173 | 0.245 | n/a |
| | BSP-Net [5] | 0.128 | 0.158 | 0.179 | 0.153 | 0.211 | 0.224 | 0.332 | 0.269 | 0.126 | 0.190 | 0.190 | 0.153 | 0.189 | 0.192 | 0.014 |
| | NSDN (ours) | 0.111 | 0.135 | 0.155 | 0.136 | **0.191** | 0.205 | 0.320 | 0.251 | 0.118 | 0.177 | **0.167** | 0.110 | 0.174 | **0.173** | 0.014 |
| IoU | OccNet [25] | 0.571 | 0.485 | 0.733 | 0.737 | 0.501 | 0.471 | 0.371 | 0.647 | 0.474 | 0.680 | 0.506 | 0.720 | 0.530 | 0.571 | 0.525 |
| | OccNet* [25] | 0.591 | **0.492** | **0.750** | **0.746** | **0.530** | 0.518 | **0.400** | **0.677** | 0.480 | **0.693** | **0.542** | 0.746 | 0.547 | **0.593** | 0.529 |
| | SIF [12] | 0.530 | 0.333 | 0.648 | 0.657 | 0.389 | 0.491 | 0.260 | 0.577 | 0.463 | 0.606 | 0.372 | 0.658 | 0.502 | 0.499 | n/a |
| | CvxNet [6] | 0.598 | 0.461 | 0.709 | 0.675 | 0.491 | **0.576** | 0.311 | 0.620 | 0.515 | 0.677 | 0.473 | 0.719 | **0.552** | 0.567 | n/a |
| | BSP-Net [5] | 0.549 | 0.371 | 0.660 | 0.708 | 0.466 | 0.507 | 0.323 | 0.638 | 0.462 | 0.667 | 0.428 | 0.711 | 0.523 | 0.539 | 0.014 |
| | NSDN (ours) | **0.613** | 0.461 | 0.719 | 0.742 | 0.515 | 0.553 | 0.368 | 0.667 | **0.516** | 0.689 | 0.511 | **0.760** | 0.550 | 0.589 | 0.014 |

Table 2: Reconstruction performance on ShapeNet [32]. In the far right column (labeled as "time"), the per object average duration (in seconds) of mesh sampling is provided to indicate the time cost for producing an evaluated mesh. In contrast to the original implementation of OccNet [25], no data augmentation is performed. Accordingly, we also report the results of pretrained OccNet trained without data augmentation, denoted as OccNet*.

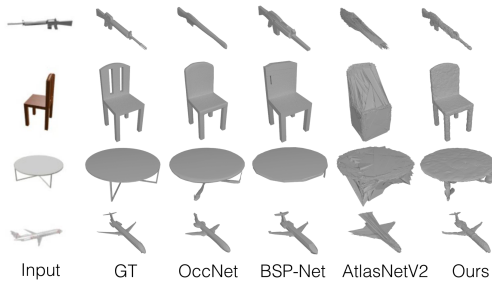

Input  GT  OccNet  BSP-Net  AtlasNetV2  Ours

Figure 3: Visualization of reconstructed meshes with an RGB image input. Best viewed when zoomed in.

| | implicit | explicit | F-score |
|---|---|---|---|
| AtlasNetV2 [7] | | ✓ | 40.05 |
| BSP-Net [5] | ✓ | | 45.60 |
| $NSDN_O$ | ✓ | | 23.93 |
| $NSDN_C$ | | ✓ | 45.84 |
| $NSDN_S$ | ✓ | ✓ | 50.52 |
| $NSDN_{S+O}$ | ✓ | ✓ | **52.27** |

Table 3: Effects of different losses on the F-score. Check marks under the implicit and explicit columns indicate whether the loss uses the corresponding shape representation. In NSDN, $O$, $C$, and $S$ indicate that only the occupancy loss, chamfer loss without surface point extraction, and surface point loss, respectively, are used.

## 4.1 Single view reconstruction

We evaluate the reconstruction performance of an NSD compared with state-of-the-art methods for an input RGB image. The quantitative results are shown in Table 2. Qualitative examples are shown in Figure 3. The number of faces of the meshes generated by NSDN and AtlasNetV2 are comparable with those by OccNet [25]. We arrive at the following conclusions: (1) NSDN consistently outperforms previous primitive-based approaches (CvxNet, SIF, BSP-Net, and AtlasNetV2) in terms of the averages of all metrics. In particular, significant improvement is observed in the F-score. (2) NSDN is relatively more effective than the leading technique (OccNet [25]), as indicated by CD1 and the F-score. It should be noted that the proposed method is comparable with OccNet, but the mesh sampling speed is distinctively faster. Details on the mesh sampling analysis can be found in Subsection 4.3.

**Effect of losses.** As the surface point loss is made available by using integrated implicit and explicit representations, we evaluate the effectiveness of the proposed loss to demonstrate the advantage of the proposed representation. We use $N = 10$ for faster NSDN training to accelerate the experiments. The results are shown in Table 3. Using only occupancy loss leads to unsuccessful training. Using the standard chamfer loss leads to performance comparable with that of previous methods. Using surface point loss outperforms leading primitive-based techniques [5]. Additionally, using occupancy loss along with surface point loss leads to slightly higher accuracy and achieves the best results.

## 4.2 Semantic capability

We evaluate the semantic capability of the proposed approach compared with other approaches based on implicit and explicit primitive representations: BSP-Net [5] and AtlasNetV2 [7]. Following the evaluation methods in [6, 5, 8], involving varying numbers of primitives for each method, we evaluate the semantic capability of the approaches as a tradeoff between representation parsimony and semantic segmentation accuracy on part labels and reconstruction accuracy measured by the F-score. For the semantic segmentation task, labels for each ground truth point are predicted as follows: (1) For each ground truth point in a training sample, we determine the nearest primitive and vote for the part label of the point, (2) we assign each primitive a part label with the highest number of votes, and (3) for each point of a test sample, we determine the nearest primitive and assign the part label of the primitive to that point. We use four classes for semantic segmentation: plane, chair, table, and lamp. For table and lamp, we follow [5] to reduce the parts from (base, pole, lampshade, canopy) $\rightarrow$ (base, pole, lampshade), and analogously for table (top, leg, support) $\rightarrow$ (top, leg). The models are trained without part label supervision.

In Figure 6, it is seen that the proposed method consistently outperforms previous methods in terms of reconstruction accuracy regardless of the number of primitives, whereas it performs comparably in the semantic segmentation task. This demonstrates its superior semantic capability. It is comparable with the method in [5] in consistent part correspondence, but it better reconstructs target shapes. The learned primitives are shown in Figure 4, where it can be seen that the proposed approach is more parsimonious in reconstructing corresponding parts.

**Effect of overlap regularization.** The high expressivity of NSD results in severe primitive overlap, leading to less interpretable part correspondence. To alleviate this, we investigate the effect of overlap regularization. As NSD is also an implicit representation, we adapt the decomposition loss proposed in [6] as an off-the-shelf overlap regularizer. The hyperparameter $\tau_r$ controls the amount of overlap. The definition of the regularizer can be found in the appendix. We set the loss weight of the regularizer to 10. In this experiment, we train the model for the airplane and chair categories. As the optimal $\tau_r$ varies across categories, we train the model with a single category. We use 1 and 1.2 for $\tau_r$ in the airplane and chair categories, respectively.

The effect of overlap regularization is shown in Figure 5. In the visualization, there is less overlap between primitives with the regularization. A quantitative evaluation is shown in Table 4. In the table, we define an overlap metric (termed "overlap"), which counts the number of 3D points inside more than one primitive. The definition can be found in the appendix. Applying the overlap regularizer clearly reduces the overlap, with a slight change in the F-score, and it improves the part IoU for both categories. In particular, the part IoU for the chair category significantly increases by 8%.

It should be noted that planar mesh patches as primitives [13, 7, 37] also have high expressivity and suffer from the same overlapping problems as NSD. Existing overlap regularization for this type of primitives, however, requires computationally expensive Jacobian computation [37]. Moreover, it is an indirect overlap regularization. We demonstrate that by simultaneously being highly expressive and an implicit representation, NSD allows for a computationally simpler and more direct approach to overcoming this shortcoming.

**Semantic part.** In Figure 1, it can be seen that a single NSD primitive (in cyan color) reconstructs the empennage. Moreover, in Figure 4, the wings (colored in green) and fuselage (colored in blue) are each reconstructed with nacelles by a single primitive. Thus, NSD can reconstruct complex shapes so that multiple parts under the same semantic part are reconstructed by one primitive. This demonstrates the expressive power of NSD in reconstructing semantic parts.

## 4.3 Mesh sampling

As the proposed method can represent surfaces in explicit forms using mesh templates, it can sample meshes significantly faster than time-consuming isosurface extraction methods. To demonstrate this, we evaluate the meshing speed and reconstruction accuracy of the proposed explicit representation compared with an implicit representation using the leading isosurface extraction method MISE [25]. For comparison, we use the same NSDN model for both representations. The results are shown in Table 5. NSD can sample meshes significantly faster than MISE with comparable F-scores (see NSDN ico#2 and MISE up#1). We also investigate the number of vertices and faces on the surface

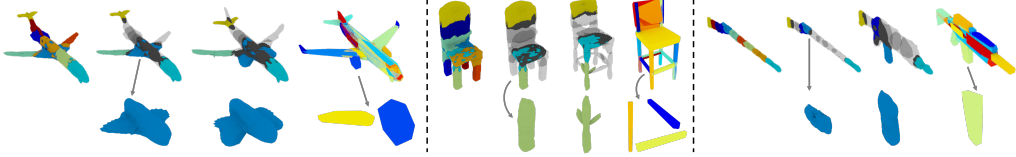

Figure 4: Primitives of different categories (plane, chair, and rifle). In each category, from far left: (1) reconstruction results with colored primitives, (2) top: only a few primitives are colored to indicate part correspondence with another reconstruction result on the right. Bottom: one primitive is selected and zoomed. (3) Top: another reconstruction result in the same category. Bottom: Same primitive as in the previous visualization. (4) Top: Reconstruction result of the same object with previous reconstruction by BSP-Net. Bottom: Manually selected primitives that correspond to the same semantic parts of the previous primitives. Best viewed zoomed in color.

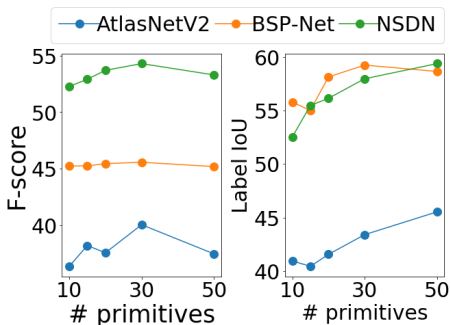

Figure 5: Effect of overlap regularization on primitive decomposition for the airplane and chair categories.

|  | airplane | | |
|---|---|---|---|
|  | Overlap | F-score | Label IoU |
| w/o reg. | 5.810 | 69.55 | 48.15 |
| w/ reg. | **0.445** | **69.92** | **50.90** |
|  | chair | | |
|  | Overlap | F-score | Label IoU |
| w/o reg. | 51.21 | **35.56** | 56.12 |
| w/ reg. | **1.16** | 33.92 | **64.37** |

Table 4: Effects of overlap regularization. The overlap score is scaled by a value of 1000 from the original value.

over mesh sampling speeds. The proposed method can produce higher-resolution meshes significantly faster than MISE. We also use the mesh sampling speed of BSP-Net [5] as a reference for implicit representation approaches with fast mesh sampling. The proposed method is comparable with that in [5]. It should be noted that we use the result of BSP-Net only in relation to meshing speed and quality, as this method focuses on low polymesh.

## 5 Conclusion

In this study, we proposed NSD as a novel primitive representation. We demonstrated that the proposed method consistently outperforms previous primitive-based approaches and that it is the only primitive-based approach performing better than the leading reconstruction technique (OccNet [25]) in a single-view reconstruction task. Moreover, it has significantly better semantic capability. In future work, we would like to integrate texture reconstruction to extend the proposed primitive-based approach to more semantic part reconstruction.

Figure 6: F-score and label IoU with varying number of primitives. The number of evaluated primitives is: 10, 15, 20, 30, and 50.

|  | #V | #F | F-score | time |
|---|---|---|---|---|
| NSDN ico0 | 2 | 5 | 34.02 | 0.012 |
| NSDN ico2 | 30 | 88 | 42.87 | 0.013 |
| NSDN ico4 | 478 | 1414 | 55.66 | 0.017 |
| MISE up0 | 12 | 31 | 26.46 | 0.051 |
| MISE up1 | 54 | 143 | 40.37 | 0.635 |
| MISE up2 | 220 | 592 | 50.28 | 5.438 |
| BSP-Net [5] | 10 | 18 | 45.60 | 0.014 |

Table 5: Mesh sampling speed for given mesh properties. $\#V$ and $\#F$ denote the number of mesh vertices ($\times 100$) and mesh faces ($\times 100$), respectively. Ico# denotes the number of icosphere subdivisions used as the mesh template of the primitive. Up# denotes the number of upsampling steps in MISE [25]. Up0 is equal to $32^3$ voxel sampling, and up2 to $128^3$.

**Broader impact**

A potential risk of NSD is that it may be used to plagiarize 3D objects, such as furniture and appliance design. As the proposed NSDN consists solely of simple fully connected layers and mesh extraction processes, it is fast and cost effective; one may be able to run this model on mobile devices with proper hardware optimization. This opens up more democratized 3D reconstruction, but it also carries the possible risk of being applied to plagiarize the design of real-world products by using 3D printers.

**Acknowledgments**

We would like to thank Antonio Tejero de Pablos, Hirofumi Seo, Hiroharu Kato, James Daniel Borg, Kenzo-Ignacio Lobos-Tsunekawa, Ryohei Shimizu, Shunya Wakasugi, Yusuke Kurose, and Yusuke Mori for their insightful feedback. We also appreciate the members of the Machine Intelligence Laboratory for constructive discussion during the research meetings. This work was partially supported by the JST AIP Acceleration Research Grant Number JPMJCR20U3, and partially supported by JSPS KAKENHI Grant Number JP19H01115.

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
