[Supplementary Material]

# Supplementary Material: Neural Star Domain as Primitive Representation

**Yuki Kawana[1], Yusuke Mukuta[1,2], Tatsuya Harada[1,2]**
[1]The University of Tokyo, [2]RIKEN AIP
{kawana, mukuta, harada}@mi.t.u-tokyo.ac.jp

## A    Example of cartesian spherical harmonics

The spherical harmonics expansion $f_\infty$ with Cartesian spherical harmonics $Y_{l,m}$ is written as follows:

$$f_\infty(\mathbf{d}) = \sum_{l=0}^{\infty} \sum_{m=-l}^{l} c_{l,m} Y_{l,m}(\omega(\mathbf{d})), \ \omega(\mathbf{d}) = (\sin\theta\cos\phi, \sin\theta\sin\phi, \cos\theta), \quad (1)$$

where $\mathbf{d} = (\theta, \phi) \in \mathbb{S}^2$, $c_{l,m} \in \mathbb{R}$ is a constant. Examples of $Y_{l,m}$ given $(l, m)$ are shown below:

$$Y_{0,0}(x,y,z) = \frac{1}{2}\sqrt{\frac{1}{\pi}}$$

$$Y_{1,-1}(x,y,z) = \sqrt{\frac{3}{4\pi}}y$$

$$Y_{1,0}(x,y,z) = \sqrt{\frac{3}{4\pi}}z$$

$$Y_{1,1}(x,y,z) = \sqrt{\frac{3}{4\pi}}x$$

$$Y_{2,-2}(x,y,z) = \frac{1}{2}\sqrt{\frac{15}{\pi}}xy \qquad\qquad Y_{2,-1}(x,y,z) = \frac{1}{2}\sqrt{\frac{15}{\pi}}yz$$

$$Y_{2,0}(x,y,z) = \frac{1}{4}\sqrt{\frac{5}{\pi}}(-x^2 - y^2 + 2z^2)$$

$$Y_{2,1}(x,y,z) = \frac{1}{2}\sqrt{\frac{15}{\pi}}zx \qquad\qquad Y_{2,2}(x,y,z) = \frac{1}{4}\sqrt{\frac{15}{\pi}}(x^2 - y^2)$$

## B    Definition of $G$ and $G^{-1}$

We define the conversion from Cartesian coordinates to the surface of the sphere $G : \mathbb{R}^3 \to \mathbb{S}^2$ as

$$G(x,y,z) = (\arctan\frac{y}{x}, \arctan\frac{\sqrt{x^2+y^2}}{z^2}) \quad (2)$$

We define the conversion from spherical coordinates to Cartesian coordinates $G^{-1} : \mathbb{R} \times \mathbb{S}^2 \to \mathbb{R}^3$ as

$$G^{-1}(r,\theta,\phi) = (r\sin\theta\cos\phi, r\sin\theta\sin\phi, r\cos\theta). \quad (3)$$

## C    Visualization of single view reconstruction

We provide an additional visualization of the single-view reconstruction results of the rifle, airplane, chair, and table categories from ShapeNet [1] in Figure 1.

Figure 1: The additional visualization of the single view reconstruction results.

## D  Visualization of primitives

We provide an additional visualization of the primitives in Figures 2, 3, and 4 for the plane, chair, and rifle categories from ShapeNet [1], respectively.

## E  Visualization of differentiable shape and surface representations

NSD provides multiple *differentiable* shape and surface representations that are available both during training and inference: mesh, surface points, normal, indicator function (signed distance function), and texture. They are visualized in Figure 5.

**Normal estimation**  As shown in Figure 5, NSD can also estimate differentiable normal vectors. Unlike methods using mesh templates, the proposed approach can derive normals at arbitrary resolution. Following [2], we derive the surface normal of the $i$th primitive $\hat{n}_i$ can be derived:

$$\hat{n}_i(\hat{\mathbf{p}}; \mathbf{t}_i) = -\frac{\partial \hat{O}_i(\hat{\mathbf{p}}; \mathbf{t}_i)}{\partial \hat{p}}, \tag{4}$$

where $\hat{\mathbf{p}} \in \hat{P}_i$ is the predicted surface point, $\hat{O}_i$ is the indicator function, and $\mathbf{t}_i$ is the translation vector of the $i$th primitive. Collective surface normal vectors $\hat{n}$ can be defined as follows:

$$\hat{n} = \bigcup_i \{\hat{n}_i(\hat{\mathbf{p}}; \mathbf{t}_i) | \forall j \in [N \setminus i], \hat{O}_j(\hat{P}_i(\mathbf{d}; \mathbf{t}_i); \mathbf{t}_i) < \tau_s, \mathbf{d} \in \{\mathbf{d}_k\}_{k=1}^K\}, \tag{5}$$

where $N$ is the number of primitives, and $\tau_s$ is a hyperparameter for the threshold of the isosurface indicator value. It should be noted that differentiable normal estimation during training by the above approach is possible through the implicit and explicit representations, whereas the approach in [2] can extract normal vectors only at inference time.

GT      Prediction      A      B      C

Figure 2: Additional visualization of the primitives for the airplane category.

GT      Prediction      A      B      C

Figure 3: Additional visualization of the primitives for the chair category.

GT  Prediction  A  B  C

Figure 4: Additional visualization of the primitives for the rifle category.

Mesh  Surface points  Indicator function

Normal  Texture

Figure 5: Differentiable shape and surface representations of NSD.

|  | mean | std |
|---|---|---|
| Superquadrics [4] | 0.042 | 0.030 |
| BSP-Net (convex) [3] | 0.070 | 0.344 |
| Proposed (star domain) | **0.154** | **0.351** |

Table 1: Mean and standard deviation of discrete Gaussian curvature [5].

Figure 6: Randomly sampled primitives: superquadrics [4], convex [3], and proposed (star domain).

## F   Analysis on expressive power of primitive shapes

We quantitatively evaluate the expressive power of NSD compared with other primitives in previous studies: convexes [3] and superquadrics [4]. We evaluate the expressive power by measuring the complexity of the inferred primitive shapes. To quantify the complexity of the shape, we evaluate the discrete Gaussian curvature [5]. We use the airplane and the chair categories from ShapeNet [1] in this evaluation. For NSD, we use $N = 10$ for the number of primitives. The mean and standard deviation of the curvature measure are shown in Table 1. A larger mean value indicates that primitive shapes have more complex surfaces in terms of unevenness, and a larger standard deviation indicates that primitives have more diverse shapes. It can be seen that NSD has larger mean and standard deviation than the methods in previous studies. This quantitatively demonstates that NSD has more expressive power, as it learns more complex and diverse primitive shapes. Randomly sampled primitives from the airplane and chair categories are shown in Figure 6.

## G   Definition of the overlap regularizer

We adapt the decomposition loss proposed in [6] as an off-the-shelf overlap regularizer. We note that we use the L1 norm instead of the L2 norm:

$$L_{\text{decomp}}(\Theta) = \mathbb{E}_{\mathbf{x} \sim \mathbb{R}^3} |\text{ReLU}(\sum_i \hat{O}_i(\mathbf{x}; \mathbf{t}_i) - \tau_r)|, \tag{6}$$

where $\tau_r$ is a hyperparameter that controls the amount of overlap.

## H   Formulation of the overlap count

We quantify primitive overlap by counting the number of 3D points inside more than one primitive as follows:

$$\text{Overlap} = \mathbb{E}_{\mathbf{x} \sim \mathbb{R}^3} \mathbb{1}(\sum_i \mathbb{1}(\hat{O}_i(\mathbf{x}; \mathbf{t}_i) \geq \tau_s) > 1). \tag{7}$$

$\mathbb{1}$ is an indicator function.