[Reviews · NeurIPS 2020]

Review 1

Summary and Contributions: This work introduces a new representation for 3D shapes based on primitives defined on a starred domain. Each primitive is represented by a unified explicit-implicit surface representation, allowing for the representation of the overall shape a simple union of primitives while guaranteeing fast surface inference.

Strengths: -the unified explicit-implicit surface representation allows for to improve both reconstruction accuracy and surface extraction performance - with respect to previous attempts to learn shape as a collection of primitives, the proposed method can handle non convex component while being accurate

Weaknesses: - I found parts of the proposed method hard to understand (see detailed comments below) - How does the proposed method deal with complicated topologies? all the visualisation provided in the paper contain genus-0 shapes. In the supplementary (Fig.3) results with different genus are provided, however, it seems like the proposed parameterisation struggles at modelling high frequency components. I would have expected a "localised" approach to be more suited for such high frequency shape details. The rebuttal addresses my concerns, so I have updated my recommendation consequently.

Correctness: Yes

Clarity: I found the paper very hard to follow and to contain some typos.

Relation to Prior Work: Yes

Reproducibility: Yes

Additional Feedback: Section 3.1: From this section, it's unclear if we are dealing only with a separate occupancy function for each point P_i, if there i a global function O (it doesn't seem so from the rest of the paper) and what P_i represent. On line 94: ... surface points \hat P which can be decomposed into a collection of N primitives. Can the authors explain more clearly the problem setting? Typos: line 2: Accurate line 95: i-th line 139: Neural -------------------------- the rebuttal addressees most of my concerns, hence I have upgraded my initial recommendation.


Review 2

Summary and Contributions: This paper tackles the task of representing an input shape as a composition of primitives and introduces a novel primitive representation (neural star domain [NSD]) towards this goal. An NSD primitive is instantiated via a (predicted) distance to surface in each direction i.e. each direction u is associated with a scalar r(u) that determines the surface position in that direction. Given an input image, this work presents an approach to predict the shape and position of a fixed number of NSD primitives s.t. their composition is similar to the underlying 3D shape.

Strengths: - I really like some aspects of the proposed primitive representation. In particular, I am very excited about the fact that both implicit and explicit computations are simple i.e. one can both a) analytically compute the occupancy of a query point w.r.t a given primitive, and b) analytically (and differentiably!) sample points on the primitive surface. While similar properties are true of simpler shape primitives e.g. cuboids/superquadrics, it is really nice to see this behavior in a more expressive representation. Just to clarify why I find this interesting, I would note that alternate representations e.g. in CVXNet, SIFs, DSIFs make the computation a) simple, but not b) i.e. sampling surface points. - The empirical results and ablations are convincing. On a note related to the above, Table 3 demonstrates that both the implicit and explicit aspects of the primitive representation are useful for designing the loss used for training. Similarly, the ablations in Table 4 are helpful and emphasize the efficiency of the approach. Finally, the empirical results show that the proposed approach is able to well-represent the underlying shapes in comparison to alternate primitives. - From a technical perspective, the idea of predicting the coefficients of the spherical harmonic functions is interesting, elegant, and novel (in this context). That said, I am curious if an implict network that, conditioned on a latent variable capturing the primitive shape, maps an input direction to a radius would be better?

Weaknesses: - While I really like the NSD representation, I am not convinced they are 'parsimoious' in a meaningful way e.g. an entire car 3D shape is an NSD, all convex shapes are NSDs. Essentially, I am concerned the space of `primitives' is too powerful ensure that its elements are simple. For example, both the primitives shown in Fig 2 are actually very complex! While this is not an issue if our only goal is to represent shapes accurately, the resulting parsing maybe undesirable for other applications e.g. shape editing. - I find the 'composite indicator function' definition (Eq 5) to be quite adhoc and actually counter-intuitive. In particular, \hat{O}_i is itself output of a sigmoid (let's say \hat{O}_i = sigmoid(V_i)). Under current definition, where \hat{O} = sigmoid(sum_i \hat{O}_i), for \hat{O} to be close to 1, the point x would need to be inside many primitives, and not just 1 - this is undesirable and encourages overlapping primitives (and is apparent from visualizations). Instead of summing \hat{O}_i, why not simply use \hat{O} = sigmoid(sum_i V_i), so if the point x is clearly inside even 1 primitive, O would be close to 1? - I assume the sentiment that Eq 6 is trying to convey is that if a point sampled from primitive i's surface is in interior of any other primitive (say primitive j), then discard it. However, the equation as written does not do this. Instead, it says that: Considering all j one at a time, if a point sampled on i's surface is in interior on this specific primitive j (not 'any' primitive j), then discard it. As the equation is currently written, a point on primitive i will be discarded only if it is inside ALL other primitives, not just any other primitive - I would guess that this is not what the paper intends and would encourage double-checking this.

Correctness: The technical details and empirical setup seem correct.

Clarity: The paper is generally well written and easy to follow.

Relation to Prior Work: I think the relation to prior work is presented accurately.

Reproducibility: Yes

Additional Feedback: I really like the proposed primitive representations and feel the approach is technically interesting and novel. In particular, I am excited about the implicit and explicit computation being feasible for this primitive representation. While I do feel the proposed primitives are not really simple or the obtained decompositions semantically too meaningful/useful, I think the technical merits of the work outweigh these and I'd recommend accepting in the hope that this or representations can be more broadly used.


Review 3

Summary and Contributions: This paper proposes to represent 3D shapes as a union of star-shaped primitives (i.e., primitive shapes where any point can be reached from some "center" via a straight line without intersections). These star-shaped neural primitives, provide a good approximation to implicit functions, maybe with slightly higher accuracy than existing solutions (e.g., convexes from BSP-net).

Strengths: This paper offers a novel shape representation for implicit functions that (a) provides a more accurate representation of the signal, (b) makes it easier to reconstruct the 3D surface.

Weaknesses: The paper mostly builds on existing ideas (i.e., OccupancyNet, BSP-NET), but I think it still makes sufficient contribution to make it an interesting work for NeurIPS audience.

Correctness: Yes

Clarity: Yes

Relation to Prior Work: OK

Reproducibility: Yes

Additional Feedback:

[Author Response · NeurIPS 2020]



Figure 1: Reconstruction of a hole with varying # of primitives $N$.

Figure 2: Reconstruction of a chair with holes with $N = 10$ primitives.

Figure 3: Reconstruction of a rifle.

We thank the reviewers for their thoughtful feedback. We are encouraged by the reviewers having identified our work to
be novel (R2, R3), easy to follow (R2), well written (R2, R3), convincing in experiments (R2), and sufficient contribution
to be an interesting work for the NeurIPS audience (R3). We are glad that they found our novel implicit-explicit
primitive representation to be very exciting, technically interesting, and elegant (R2), contributing to more accurate
shape reconstruction (R1, R2, R3) and improving surface reconstruction performance (R1, R2, R3). We address the
reviewers' comments below.

R1 **Can the authors explain more clearly the problem setting?:** We believe our work is easy to understand overall,
given that R2 and R3 evaluated our paper as "easy to follow" and "well written." We agree that there are some confusing
points, which we will address in the camera ready. Given this, we would like to clarify the problem setting once again.
The goal of this research is to learn a model that accurately reconstructs the target shape characterized by an indicator
function $O$ (L92) and a set of surface points $P$ (L91), by predicting the corresponding approximation $\hat{O}$ and $\hat{P}$ (L96).
Moreover, to understand the target shape structure, we reconstruct the shape by combining multiple semantic parts
(primitives). To do so, we define a primitive characterized by an indicator function $\hat{O}_i$ and a surface point function $\hat{P}_i$
(Section 3.3). Note that $\hat{P}$ is a set, $\hat{P}_i$ is a function, and $\hat{P}_i(\mathbb{S}^2)$ is a set of surface points of a primitive. We also study
how to combine $\hat{O}_i$ and $\hat{P}_i$ to represent $\hat{O}$ and $\hat{P}$ (Section 3.4). In the camera ready, we will address the confusing
notation in Eq. 6 and L97, in which $\hat{P}$ takes arguments like a function, although it is a set.

R1 **How does the proposed method deal with complicated topologies?:** By increasing the number of primitives
$N$, our model learns to handle complex topologies such as holes (see Figure 1). Note that even with a small number of
primitives (parsimony is an essential criterion in the primitive based approaches), our approach can handle complicated
topologies better than the leading method (BSP-Net), as shown in Figure 2. Although small holes are difficult to deal
with, other high-frequency details such as small parts are reconstructed better by our approach. For example, in Figure
3, our method successfully reconstructs the rifle's three distinct handles while other methods fail. Our method works
better because the explicit surface of NSD enables the optimization of shapes directly against the points sampled from
the small parts, while implicit based methods tend to miss such small parts during sampling and training.

R3 **The paper mostly builds on existing ideas:** We agree that we strongly build our method in existing ideas, but
we have developed on these ideas and made novel progress and several contributions. First, we propose a novel,
**differentiable** implicit-explicit representation. BSP-Net realized the instant surface extraction during inference, but
it needs a complex surface approximation scheme during training. We take a step further to realize the exact and
differentiable surface extraction in a simple and novel manner (as R2 agrees), improving the reconstruction accuracy
(Table 3 in the paper). Moreover, our proposed primitive representation is far more expressive than previous works.
(see Figure 1 in the paper). Although previous works have gradually improved the primitives' expressivity, their
low-dimensional parameter space still limits it. We propose NSD, whose expressivity is equivalent to a capacity of
neural network (see supplementary Section B for proof), realizing far more expressive primitives. We believe these
novel contributions make our work sufficient to be a good conference paper.

R2 **Too expressive primitive representation leads to less meaningful part decomposition:** We appreciate R2 for
raising the concern around the critical question: how we should evaluate the quality of the decomposition result under
self-supervised settings. Having the same concern, and following the previous works (BSP-Net, CVXNet), we evaluated
our work based on the consistency with parts annotated by humans because we would like to know how meaningful the
decomposition result is for **humans**. In Figure 5 in the paper, we show that the part decomposition of our method is
semantically consistent with human annotations, comparable to the leading method in this task (BSP-Net).

R2 **Current composite indicator function $\hat{O}$ unfavorably encourages the overlapping primitives:** We appreciate
R2's constructive suggestion; we also had the same concern. Actually, we considered $\mathrm{Sigmoid}(\sum_i V_i)$. However, as
the region of $V_i$ includes both positive and negative domains, the summation can unfavorably cancel out each other
terms. We tried ReLU instead of sigmoid for less overlap in Eq. 3, but we experimentally found sigmoid works only
slightly better in terms of overlap by 6%. In the camera ready, we will report the overlap regularizer result.

R2 **Eq. 6 needs the double-checking:** We appreciate R2 again for pointing this out. we will fix the Eq. 6 in the
camera ready as follows: $\hat{P} = \bigcup_i \{\hat{P}_i(\mathbf{d}; \mathbf{t}_i) | \forall j \in [N \setminus i], \hat{O}_j(\hat{P}_i(\mathbf{d}; \mathbf{t}_i); \mathbf{t}_i) < \tau_s, \mathbf{d} \in \{\mathbf{d}_k\}_{k=1}^K\}$.

[Meta-Review · NeurIPS 2020]

This work introduces a new representation for 3D shapes based on primitives defined on a starred domain. Reviewers find the proposed approach novel and the empirical results convincing. Reviewers brought up some weaknesses mostly related to clarity, which were sufficiently addressed by the rebuttal. All reviewers recommend accept after the rebuttal. AC did not find sufficient grounds to overturn this consensus recommendation. The authors should revise the paper by incorporating the discussion in the rebuttal.